# The Need for a Specialized Neurocognitive Screen and Consistent Cognitive Impairment Criteria in Spinal Cord Injury: Analysis of the Suitability of the Neuropsychiatry Unit Cognitive Assessment Tool

**DOI:** 10.3390/jcm11123344

**Published:** 2022-06-10

**Authors:** Danielle Sandalic, Yvonne Tran, Ashley Craig, Mohit Arora, Ilaria Pozzato, Grahame Simpson, Bamini Gopinath, Jasbeer Kaur, Sachin Shetty, Gerard Weber, Lisa Benad, James W. Middleton

**Affiliations:** 1John Walsh Centre Rehabilitation Research, The Kolling Institute, Royal North Shore Hospital, St Leonards, NSW 2065, Australia; danielle.sandalic@health.nsw.gov.au (D.S.); mohit.arora@sydney.edu.au (M.A.); ilaria.pozzato@sydney.edu.au (I.P.); grahame.simpson@health.nsw.gov.au (G.S.); james.middleton@sydney.edu.au (J.W.M.); 2Faculty of Medicine and Health, Sydney Medical School, The University of Sydney, Sydney, NSW 2006, Australia; 3SCI Unit, Royal North Shore Hospital, St Leonards, NSW 2065, Australia; jasbeer.kaur@health.nsw.gov.au (J.K.); lisa.benad@health.nsw.gov.au (L.B.); 4Australian Institute of Health Innovation, Macquarie University, North Ryde, NSW 2113, Australia; yvonne.tran@mq.edu.au (Y.T.); bamini.gopinath@mq.edu.au (B.G.); 5SCI Unit, Prince of Wales Hospital, Randwick, NSW 2031, Australia; sachin.shetty@health.nsw.gov.au; 6SCI Unit, Royal Rehab, Ryde, NSW 2112, Australia; gerard.weber@royalrehab.com.au; 7Spinal Outreach Service, Royal Rehab, Ryde, NSW 2112, Australia

**Keywords:** spinal cord injury, neurocognitive function, mild cognitive impairment, executive function, memory, attention, processing speed

## Abstract

The assessment of mild cognitive impairment (MCI) following spinal cord injury (SCI) is vital. However, there are no neurocognitive screens which have been developed specifically to meet the unique requirements for SCI, nor are there consistent MCI criteria applied to determine the rates of MCI. The aim of this study was to determine the suitability of a neurocognitive screen for assessing MCI in adults with SCI. A total of 127 participants were recruited. Socio-demographic and injury related variables were assessed. All participants completed the screen. Descriptive statistics are provided for total/domain screen scores and all items, and the screen’s ability to distinguish MCI was examined. Congeneric confirmatory factor analyses (CFA) were employed to investigate structural validity. The screen total score was sensitive to differences in neurocognitive capacity, as well as for time since the injury occurred (*p* < 0.01). The MCI rate ranged between 17–36%. CFA revealed attention and visuoconstruction domains had an adequate model fit and executive function had poor fit, while CFA models for memory and language did not fit the data (did not converge), hence could not be determined. While the screen differentiated between those with MCI and those without, and MCI as a function of time since injury, limitations of its suitability for assessing MCI after SCI exist, demonstrating the need for a specialized neurocognitive screen for adults with SCI.

## 1. Introduction

Mild to moderate cognitive impairment (MCI) is thought to be a significant comorbidity of spinal cord injury (SCI) [1,2,3,4,5], with studies reporting prevalence rates of MCI between 10–60% [1,2]. Recent research suggests that adults with SCI are 13 times more susceptible to experiencing MCI compared to able-bodied individuals [2]. Time since injury has been highlighted as a factor influencing MCI rates, with research suggesting that the severity of MCI worsens over time, akin to an accelerated cognitive aging process [4]. Nevertheless, reliable MCI rates are not yet available, due to factors such as the heterogeneity of samples, differences in criteria used to define MCI, numerous neurocognitive tests used to assess MCI, time since injury, and a failure to distinguish between sub-types of MCI, such as deficits in attention versus deficits in memory [1,2,5,6].

Multiple factors are thought to contribute to MCI in adults with SCI [1,2]. Comorbid or pre-existing mild to severe traumatic brain injury (TBI) is considered to be a common cause of cognitive impairment in adults with SCI [1,2,5,7,8]. However, the contribution of TBI requires clarification, as critical markers for diagnosing mild TBI, such as post-traumatic amnesia, can rapidly resolve post-injury, and are inconsistently assessed in emergency/acute care when managing SCI. Therefore, the occurrence of mild TBI following SCI is most likely underdiagnosed [9]. Other possible contributors to MCI include psychological morbidities, such as depression, older/advancing age, fatigue, polypharmacy, alcohol/substance abuse, nervous system inflammation associated with SCI, and disordered autonomic/cardiovascular control [1,2,5,10,11,12,13,14]. Given the wide variety of mechanisms leading to MCI after SCI, it is difficult to use any one clinical indicator to detect MCI. Therefore, a statistical approach, based on test performance, offers a useful alternative [2,5].

Rehabilitation following SCI involves intensive learning and employment of new skills to prevent complications and improve functionality and adjustment, and, arguably, successful outcomes will be considerably hindered by deficits in cognitive function, such as impairments in memory, executive function, attention, and visuoconstruction ability [1,2]. It is therefore crucial that neurocognitive assessment/screening be performed universally following a SCI, especially during intensive rehabilitation. Further, neurocognitive measures employed to assess MCI should be investigated for their psychometric properties when applied to the SCI population. This should result in more reliable assessment of MCI for a person with SCI, improving clarity regarding cognitive capacity status, and hopefully leading to more directed therapeutic strategies for those with co-morbid MCI [1,2]. Unfortunately, there has been very limited research focused on the validity of neurocognitive tests used for the assessment of MCI in the SCI population, and the authors are unaware of any neurocognitive screen developed specifically for SCI [2,5].

A validated neurocognitive screen called the Neuropsychiatry Unit Cognitive Assessment tool (NUCOG) has been shown to be a reliable and sensitive screen to assess MCI in patients with disorders such as Alzheimer’s Disease, neurological disorders without dementia, and psychiatric disorders, such as depression and psychoses [15]. Recent research into cognitive capacity following SCI used the NUCOG, with probable MCI defined as one standard deviation below the NUCOG norm for a sample of able-bodied adults [2,15]. Findings demonstrated that the NUCOG has clinical utility in a SCI population, and that it could be used reliably for referral for comprehensive neurocognitive assessment if probable MCI was detected [2]. Those with cognitive impairment were also shown to be at higher risk of clinically elevated depressive mood following discharge from rehabilitation [2]. SCI NUCOG norms are available [16]. The NUCOG is based on comprehensive neurocognitive tests such as the Weschler Adult Intelligence Scale (WAIS), which have demonstrated factor structure in non-SCI disordered populations [17]; however, the five cognitive domain factor structure of the NUCOG (attention, executive, language, memory, and visuoconstruction) has not yet been validated in a SCI population. Therefore, the aim of this paper included an investigation into the suitability of the NUCOG as a neurocognitive screen for adults with SCI by examining its sensitivity in detecting cognitive impairment in SCI, especially as a function of time since injury. An additional aim included the exploration of MCI criteria applied to SCI, as well as an examination of the structural validity of the NUCOG domains using congeneric confirmatory factor analysis (CFA).

## 2. Materials and Methods

### 2.1. Participants

Participants included adults with SCI who were either engaged in intensive rehabilitation (*n* = 97) or living in the community following discharge from rehabilitation (*n* = 30). The recruitment procedure has been described in detail elsewhere [2,10]. Rehabilitation units included the three SCI Units in Sydney, NSW, Australia. All three units have similar medical, physical, and psychosocial SCI rehabilitation programs. Participants were recruited using an opt-in approach, both for those in the SCI Units and those in the community. Inclusion criteria consisted of: (a) the presence of an acute or chronic SCI; (b) a recent first-time admission to a SCI unit, a re-admission as an inpatient, attending a SCI outpatient clinic, or living in the community; (c) aged 18–80 years at the time of interview, and (d) English language proficiency. Exclusion criteria included (a) the presence of severe cognitive impairment (e.g., severe TBI, loss of consciousness >24 h, and Glasgow Coma Scale of <9) and (b) severe psychiatric disorder (e.g., florid schizophrenia or bipolar disorder) that prevented the person from participating and completing the NUCOG. Full compliance with the Code of Ethics of the World Medical Association occurred, and the local institutional human research ethics committee granted ethics approval. All participants provided informed consent prior to participating.

### 2.2. Study Design and Procedure

Information presented in this paper formed part of two prospective group cohort studies, the results of which have been published [2,10,18]. Data were taken from one time point for both groups. Health professionals trained in using the NUCOG assessed all participants either in hospital or community settings. Participants completed the NUCOG only once, as well as a comprehensive assessment, including psychological, socio-demographic, and injury-related measures. Only socio-demographic, injury, and NUCOG data are presented in this paper. Social distancing was practiced, and personal protective equipment was worn when testing recently recruited participants to reduce the risk of potential COVID-19 infection.

### 2.3. Measures

Socio-demographic data were obtained from interview and/or medical records. Medical specialists assessed the level and extent (completeness) of lesions in compliance with the International Standards for Neurological Classification of SCI (http://ais.emsci.org/, accessed on 6 June 2016). Cognitive capacity was assessed using the NUCOG [15], consisting of 21 items that assess cognitive function across five domains: attention, executive function, language, memory, and perceptual/visuoconstruction. Senior neuropsychologists were involved in its development, and it is based on multiple neurocognitive tests such as the Stroop, Trail Making Test, and WAIS-4th Edition [15,17]. NUCOG has demonstrated criterion, convergent, and discriminant validity (e.g., between SCI and able-bodied samples), as well as acceptable reliability and specificity/sensitivity [15,16]. As a neurocognitive screen, it can be administered in approximately 30–40 min, providing a total score of 100, with each of the five domains having a total score of 20, with higher scores indicating greater cognitive capacity. Norms for the general population, stroke, head injury, epilepsy, Alzheimer’s disease, psychiatric disorder [15], and SCI [16] are available.

Some tasks of the NUCOG require hand motor skills that can be impaired in people with physical disabilities such as tetraplegia [2]. Therefore, if the full test is to be used, alternative means of assessing these cognitive tasks are necessary, as per standard practice in cognitive testing of people with physical disabilities [19]. Research in adults with limited hand function relies on motor-free neuropsychological assessment [2,19], with items requiring hand motor capacity being eliminated, such as drawing a clock face with a specific time of the day. However, eliminating such items may lead to a loss of information in the test, while adapting/modifying items may possibly provide different cognitive domain data.

For this study, it was decided that all NUCOG items would be retained when assessing individuals with high-level spinal lesions, through the provision of alternative instructions. Based on prior research [2], it was concluded that these modifications tapped cognitive function close to the original purpose of those items. For instance, in a visuoconstruction reproduction task, participants with limited hand function were asked to describe in detail the shapes, rather than draw them with their hands. The prior research involved a comparison of the NUCOG domain scores for paraplegia versus tetraplegia (hands-free). No significant differences were found across the five domains, with, for example, <0.15 mean difference in scores out of a total 20 for each of the visuoconstruction and executive domains [2].

### 2.4. Statistical Analysis

Central tendency statistics were generated for NUCOG total score, domains, and items. A multivariate analysis of covariance (MANCOVA) was conducted to determine differences in the total NUCOG score due to time since injury (those assessed in rehabilitation versus those in the community), with age and sex entered as covariates. Factors complied with rules governing normality [20]. Based on a conservative effect size of 0.25 (Cohen’s f), using MANCOVA, an α probability of 0.05 and a minimum sample size of 112 was required for “a priori” statistical power to be >80%. An exploratory Pearson correlation analysis was also conducted to determine associations between the NUCOG domains and socio-demographic and injury-related factors.

Insufficient statistical power was available to conduct a 5-factor CFA on the NUCOG with 21 items. Consequently, five congeneric (one-factor) CFAs were conducted, using a maximum likelihood estimator that was robust to the non-normality and non-independence of observations. Congeneric CFA assumes that the covariance among items is due to a single common factor. Maximum likelihood estimation with a robust variance estimator (Huber–White) was used with a scaled test statistic (asymptotically) equal to the Yuan–Bentler test statistic. Each congeneric CFA involved around 12 parameters, and with 10 participants per parameter, we required a maximum of 120 persons to achieve acceptable power [21]. Fit statistics were used to evaluate the suitability of each domain, consistent with the CFA literature [22], that is, (i) non-significant chi-square (χ^2^); (ii) values of ≥0.90 for the comparative fit index (CFI) and Tucker–Lewis index (TLI); (iii) ≤0.06 for the Standardized Root Mean Square Error of Approximation (RMSEA); and (iv) <0.08 for the Standardized Root Mean Residual (SRMR) [23]. Mplus 7.3 (http://www.statmodel.com/verhistory.shtml, accessed on 6 June 2016) was used for the CFA.

## 3. Results

Socio-demographic and descriptive statistics for NUCOG total scores, domain scores, and rehabilitation (*n* = 97; mean 0.15 years since injury) versus community (*n* = 30; mean 6 years since injury) sub-groups are shown in Table 1. As was perhaps expected, those assessed when living in the community had a higher mean age and more years having passed since the injury (*p* < 0.01). Using MANCOVA with age and sex as covariates (Wilks Lambda = 0.82, F_5119_ = 5.00, *p* < 0.001), those living in the community had significantly higher NUCOG total and domain scores compared to those who were assessed when in rehabilitation.

Different criteria can be used to define MCI. The NUCOG total mean norm reported in the NUCOG manual [15] was 92.9 (SD = 4.9). Therefore, in this sample, an MCI cut-off score of 88 (93 − 5) was used, resulting in an MCI rate of 27.5% (35/127). However, if MCI is defined as 1 SD below the NUCOG total mean for the current sample, then the MCI cut-off score is ≤85 (91.25 − 6.4), and 17.3% (22/127) would be estimated to have probable MCI. For the community sample, the cut-off would be higher at ≤90 (94.41 − 4.1), and the MCI rate would be 16.6% (5/30). For those in rehabilitation, the MCI cut-off score would be ≤84 (90.27 − 6.7), and the MCI rate would be 19.6% (19/97). Table 2 shows the percentiles of cognitive capacity for the total NUCOG score for all participants. If defines MCI is defined as ≤ 90th percentile rank, then 36.2% of the sample would be estimated to have probable MCI. If MCI was defined as ≤85th percentile, then the MCI rate would be 17.3%.

Table 3 shows the means and variances for all 21 items. The items with the least response variance reflect a group of items tapping ‘basic’ cognitive processes, such as attention item 1, relating to orientation to time and space, and memory item 3, which tests declarative memory for autobiographical and semantic facts. Those items showing the most variance seemingly tap ‘higher-order’ processes, which are arguably dependent on premorbid intelligence (e.g., arithmetic skills assessed by visuoconstruction item 5, and working memory capacity assessed by attention items 2 and 2-2). Tests of intelligence might be expected to contain item variance, as they serve to differentiate individuals’ cognitive potentials; however, if the purpose of a screen is to assess ‘impairment’, it could be argued that the inclusion of high item variance could contribute to false positive identification of impairment where there is lower-than-average premorbid intelligence. Table 4 shows Pearson correlations between the domains and the socio-demographic and injury factors. As expected, all NUCOG domains were positively correlated. Stronger associations were found between attention and visuoconstruction (0.53), and between memory and executive function (0.46). There were few significant associations between demographic variables and NUCOG domains. Table 5 shows the fit indices for the congeneric CFA results for the NUCOG domains. Attention and visuoconstruction were found to have an adequate fit, while executive function had a poor fit. The maximum likelihood for memory and language did not converge, that is, it did not fit the data, and so the model parameters could not be determined. One reason for this is the lack of variability, as shown in Table 3.

## 4. Discussion

The NUCOG total score was shown to measure cognitive capacity suitably in adults with SCI. Almost two-thirds scored >90th percentile, around 30% scored between the 81–90th percentiles, and around 5% scored ≤80th percentile. We maintain that this is consistent with neurocognitive screen scores for adults with SCI, many of whom do not have deficits in cognitive capacity [2,24,25]. Additionally, NUCOG detected differences associated with time since injury. Participants in rehabilitation were found to have lower cognitive scores, as compared to community participants. Lower cognitive scores in participants engaged in rehabilitation might be explained, at least partially, by compromised attentional resources associated with anxiety, impacts of emergency and acute medical environments, frequent use of neuroleptic medications, nervous system inflammation following spinal damage, decentralized cardiovascular control, sleep disorder, depressive mood, elevated fatigue, and so on [1,2,5]. Furthermore, context- or state-dependent cognitive decline could be expected to improve alongside physical healing and improved adjustment over time. The difference between community and rehabilitation participants must be viewed cautiously, as it is based on cross-sectional data and smaller numbers in the community sample. Prior cross-sectional research has argued that MCI worsens over time for adults with SCI [4]. Further research is required to explore the influence of time since injury on MCI rates.

The importance of having standard criteria for what constitutes MCI when using a neurocognitive test/screen is highlighted by the varying diagnostic rates found based on different criteria [5]. If a traditional ≤1 SD criterion is used based on NUCOG population norms [2,15], then the MCI rate is 27%. However, if a ≤1 SD criterion below the current sample mean is used, the MCI rate lies between 17–20%. It has been argued that percentile ranks should be used to communicate MCI rates, since they communicate how common/uncommon test scores are normatively [24]. If a ≤90th percentile rank criterion is used, the MCI rate climbs to 36%. Variations in MCI rates following SCI relate to multiple factors [1,2,3,5], but clearly, the MCI criteria used may influence rates substantially. This emphasizes the importance of employing standard, universally agreed upon criteria for detecting probable MCI in SCI when using neurocognitive tests/screens. If this can be achieved, then probable MCI could be determined reliably, allowing for prompt referral for comprehensive neurocognitive assessment, as advised by the National Academy of Neuropsychology (USA) [25]. The result of an inconsistent use of criteria for detecting MCI has been demonstrated [26]. We suggest, when establishing the rate of MCI in SCI, that a standard deviation from a population or sample norm in combination with percentile ranks be considered as an alternative to criteria based solely on SD deviations from a mean [24,25].

Only two (attention and visuoconstruction) of the five NUCOG domains were found to have acceptable structural validity using congeneric CFA. The failure to support the factor structure of all five NUCOG domains means that one should be cautious if basing MCI rates on these domains. Notably, independent research concluded that MCI rates based on domains are problematic in non-SCI areas [26]. Furthermore, failure to validate the factor structure of neurocognitive screens is not uncommon, given they have fewer items and less variance, as compared to comprehensive neurocognitive tests [27]. For example, the Montreal Cognitive Assessment (MOCA) was shown to have poor domain factor structure in dementia populations [27].

The failure to find structural validity in three domains possibly highlights the existence of shared variance in cognitive scores between domains. This has been called the “g” factor, which indicates the existence of core/latent cognitive mental abilities that share variance with performance across task-specific items in domains [28]. Such a latent factor is possibly a contributor to the poor structural validity of the executive, language, and memory domains, although a major contributor must also be low item variance in some NUCOG domains. After all, cognitive tests assess tasks that are multifactorial in nature, involving motor, language, memory, visuospatial, and executive skills [28,29]. Failure to validate the structure of these three domains may also be associated with the NUCOG being designed for neuropsychiatric populations, rather than for the special needs of SCI. While the structural validity of the NUCOG domains was not fully supported, its five domains are important. The Diagnostic and Statistical Manual of Mental Disorders-5 (DSM-5) defines six key neurocognitive domains believed to be important for identifying MCI, being attention, perceptual motor function (including visuoconstruction reasoning), memory, executive function, language, and social cognition [30]. The sixth DSM-5 domain, “social cognition”, is not assessed by the NUCOG [15].

Limitations of this research include the cross-sectional nature of the data and the relatively small sample size of the community participants. To address these limitations, we are currently conducting prospective research involving the assessment of MCI soon after injury, up to 12-months post-SCI. A recent critical review highlighted some of these limitations in research in this area, and concluded that substantial heterogeneity exists when assessing MCI after SCI, due, in part, to factors such as varying times since injury, sample sizes, types of tests used, and, importantly, lack of neurocognitive tests specialized for the needs of SCI [31]. We believe that our findings support the need for the development of a neurocognitive screen, developed specifically to address this heterogeneity and the particular needs of SCI. If an adequate screen is not developed, subjective measures of cognitive capacity will continue to be used, such as the Functional Independence Measure [32], or, alternatively, a variety of objective neurocognitive tests which are not suited to SCI will be used, subject to the same limitations discussed above [5].

## 5. Conclusions

While we believe that the NUCOG total score can be used judiciously to detect probable MCI after SCI, the failure to confirm the structural validity of all NUCOG domains is a significant problem. Arguably, MCI rates following SCI will remain uncertain until a specialized neurocognitive screen is developed with research controls for factors that influence MCI rates, such as time since injury. Furthermore, even though we have employed “hands-free” alternatives for items requiring upper limb motor control [2], we cannot be certain that these alternative items are assessing the intended cognitive domain in the original NUCOG test. Therefore, we advocate for a specialized SCI hands-free test. As mentioned earlier, evidence indicates that SCI is associated with accelerated cognitive ageing, resulting in, for instance, slowed electroencephalographic brain activity and slowed processing speed [3,4,33,34,35]. Gross deficits in memory, for example, would not generally be expected, as this is typically a marker of dementia. We believe, then, that a specialized test should target cognitive domains affected by SCI, such as accelerated cognitive ageing and fluid cognitive functioning (or fluid intelligence), involving, for example, executive functioning and attention/processing speed domains [36], rather than, for example language and rote memory. Research will also need to investigate factors responsible for accelerated cognitive ageing (e.g., inflammation, sleep disorder, polypharmacy, mood disorder). It is hoped that this research on the suitability of a neurocognitive screen for SCI and the need for consistent MCI criteria will lead to improved identification and management of MCI associated with SCI, as well as improved life outcomes [37,38].

## Figures and Tables

**Table 1 jcm-11-03344-t001:** Descriptive statistics for socio-demographic and injury factors and differences between participants in rehabilitation and those in the community for the NUCOG total score and the five domains.

	Rehabilitation *n* = 97	Community *n* = 30	Combined *n* = 127
Sex males, *n* (%)	72 (74)	28 (93)	100 (79)
Age mean years (SD)	44.82 (18)	53.60 (14) *	46.90 (18)
Years education, mean (SD)	12.84 (2.5)	13.14 (2.2)	12.92 (2.4)
Tetraplegia, *n* (%)	39 (40)	11 (37)	50 (39)
Years since injury, mean (SD)	0.15 (0.1)	6.00 (6.3) *	1.54 (3.9)
Complete lesion, *n* (%)	44 (45)	21 (70)	65 (51)
NUCOG total mean (SD)	90.27 (6.7)	94.41 (4.1) *	91.25 (6.4)
Attention, mean (SD)	17.00 (2.7)	18.60 (1.9) *	17.40 (2.7)
Visuoconstruction, mean (SD)	18.71 (1.4)	19.73 (0.5) *	18.95 (1.3)
Memory, mean (SD)	17.72 (2.0)	18.43 (1.9) *	17.89 (2.0)
Language, mean (SD)	19.28 (0.9)	19.83 (0.4) *	19.41 (0.8)
Executive function, mean (SD)	17.76 (2.2)	17.82 (2.2) *	17.7 (2.2)

* *p* < 0.01 for differences between the rehabilitation and community samples.

**Table 2 jcm-11-03344-t002:** Percentiles of cognitive capacity scores for the total NUCOG score for N = 127.

NUCOG Score/100	Count	Cumulative Count	Percent	Cumulative Percent
≤75	4	4	3.14961	3.1496
>75 and ≤80	3	7	2.36220	5.5118
>80 and ≤85	15	22	11.81102	17.3228
>85 and ≤90	24	46	18.89764	36.2205
>90 and ≤95	40	86	31.49606	67.7165
>95	41	127	32.28346	100.0000

**Table 3 jcm-11-03344-t003:** Descriptive statistics and variance for all NUCOG items for the adults with SCI. Each domain has a total score of 20, with a total NUCOG score of 100.

NUCOG Item		Descriptive Statistics for All NUCOG Items					
	Possible Score	N	Mean	Median	Minimum	Maximum	SD
**a 1**	5	127	4.94	5	4	5	0.24
**a 2**	4	127	3.13	4	0	4	1.14
**a 2-2**	4	127	2.48	3	0	4	1.46
**a3**	7	127	6.83	7	0	7	0.89
**v 1**	4	127	3.90	4	2.5	4	0.29
**v 2**	4	127	3.85	4	3	4	0.31
**v 3**	4	127	3.90	4	2	4	0.37
**v 4**	4	127	3.92	4	3	4	0.26
**v 5**	4	127	3.33	4	1	4	0.93
**m 1**	3	127	3.00	3	3	3	0
**m 1-2**	3	127	2.05	2.5	0	3	0.99
**m 2**	8	127	7.14	8	0.5	8	1.42
**m 3**	6	127	5.61	6	3	6	0.65
**e 1**	4	127	3.56	4	0	4	0.96
**e 2**	10	127	8.93	10	4	10	1.62
**e 3**	4	127	3.26	3.5	0	4	0.92
**e 4**	2	127	1.94	2	0	2	0.26
**l 1**	4	126	3.90	4	3	4	0.29
**l 2**	5	127	4.91	5	3	5	0.33
**l 3**	5	127	4.95	5	3	5	0.25
**l 4**	2	127	1.64	2	0	2	0.50
**l 5**	2	127	1.97	2	0	2	0.25
**l 6**	2	127	1.99	2	1.5	2	0.04

a = attention; v = visuoconstruction; m = memory; e = executive; l = language; SD = standard deviation. Note, for m1, all participants scored maximum points, so there is no variation, thus no standard deviation was calculated.

**Table 4 jcm-11-03344-t004:** Exploratory correlation analysis between the NUCOG domains and socio-demographic and injury related factors for all participants (N = 127).

	Att	Mem	Vis	Exec	Lang	Age	Sex	TSI	Level	Yr Ed
Att	---	0.36 ***	0.53 ***	0.26 **	0.38 ***	−0.14	0.05	0.12	0.13	0.19 *
Mem		---	0.42 ***	0.46 ***	0.09	−0.25 **	0.08	−0.05	−0.03	0.14
Vis			---	0.28 **	0.36 ***	0.07	−0.05	−0.15	0.11	0.26 **
Exec				---	0.32 ***	−0.28 **	0.26 **	−0.03	0.03	0.16
Lang					---	0.02	0.11	0.18	0.03	0.09
Age						---	−0.10	0.19 *	−0.11	0.07
Sex							---	−0.17	0.06	0.07
TSI								---	−0.06	−0.05
Level									---	−0.03
Yr Ed										---

*** < 0.001 ** < 0.01 * < 0.05; Att: attention; Mem: memory; Vis: visuoconstruction; Exec: executive; Lang: Language; TSI: time since injury; Yr Ed: years of education.

**Table 5 jcm-11-03344-t005:** Fit statistics for the congeneric CFA results for the 5 NUCOG domains.

Domain	χ^2^	df	p	TLI	CFI	RMSEA (90% CI)	SRMR
**Attention**	2.67	2	0.23	0.92	0.97	0.07 (0.00, 0.28)	0.06
**Visuoconst**	6.15	5	0.29	0.85	0.92	0.05 (0.00, 0.15)	0.05
**Executive**	16.9	2	<0.01	0.31	0.77	0.12 (0.07, 0.18)	0.05
**Memory**							Not fitted
**Language**							Not fitted

df: degrees of freedom, p: probability of significance; Visuoconst: visuoconstruction. χ^2^: Chi square test: non-significant. χ^2^ outcome required for goodness of fit. TLI: Tucker–Lewis index: >0.90 for adequate fit. CFI: comparative fit index: >0.9 for adequate fit. RMSEA: root mean square error of approximation: ≤0.06 for adequate fit. SRMR: Standardized Root Mean Squared Residual: <0.08 for adequate fit.

## Data Availability

The datasets generated are available from the corresponding author upon reasonable request.

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
