# Peer review of "The Need for a Specialized Neurocognitive Screen and Consistent Cognitive Impairment Criteria in Spinal Cord Injury: Analysis of the Suitability of the Neuropsychiatry Unit Cognitive Assessment Tool"

_jcm, 2022, doi:10.3390/jcm11123344_

Round 1
Reviewer 1 Report
Sandalic and colleagues are presenting a study assessing the suitability of a neurocognitive screen (NUCOG) in detecting mild cognitive impairment (MCI) in patients suffering from spinal cord injuries. The study in itself is interesting and well-written, but the authors need to emphasize what’s different from their previous work. For example, the evaluation of NUCOG for MCI in spinal cord patients was assessed in ref. 16 and ref.2.
The result/discussion section for Table 3 is short. A few of the NUCOG items seems to be more affected than the others such as items a 2-2, , 1-2. and e3. The authors may want to discuss this.
Line 209. The authors state the Table 4 shows Pearson correlations between the domains and socio-demographic and injury factors, but then they describe only the correlations between the NUCOG domains. The authors should also describe the correlations between the domains and socio-demographic/injury factors. The authors may also want to do correlation analysis for total NUCOG score.
Line 31. In the sentence: “…and its ability to…” it is not clear what the “its” refers to. Please clarify.
Author Response
Sandalic and colleagues are presenting a study assessing the suitability of a neurocognitive screen (NUCOG) in detecting mild cognitive impairment (MCI) in patients suffering from spinal cord injuries. The study in itself is interesting and well-written, but the authors need to emphasize what’s different from their previous work. For example, the evaluation of NUCOG for MCI in spinal cord patients was assessed in ref. 16 and ref.2.
Response: Thank you for these kind comments. The material presented in this current paper is different to that presented in the two cited papers. Ref 2 reported (a) differences between adults with SCI and an able-bodied sample for the NUCOG domains. (b) it also studied the possible influence of cognitive impairment on mood during inpatient rehabilitation and when living in the community. Ref 16 is not a journal paper: it is primarily a manual for NUCOG norms for adults with SCI. We have pointed out the focus of these two references in lines 76-81 and added the following to make the distinction more clearly:
"Those with cognitive impairment were also shown to be at higher risk of clinically elevated depressive mood following discharge from rehabilitation [2]."
The result/discussion section for Table 3 is short. A few of the NUCOG items seems to be more affected than the others such as items a 2-2, , 1-2. and e3. The authors may want to discuss this.
Response: Thank you for this comment: we have added the following material to lines 210-218:
The items with least response variance reflect a group of items tapping ‘basic’ cognitive processes such as attention item 1, relating to orientation to time and space, and memory item 3, which tests declarative memory for autobiographical and semantic facts. Those items showing most variance seemingly tap ‘higher-order’ processes arguably dependent on premorbid intelligence (e.g., arithmetic skills assessed by visuoconstruction item 5 and working memory capacity assessed by attention items 2 and 2-2). Tests of intelligence might be expected to contain item variance as this serves to differentiate individuals’ cognitive potentials; however, if the purpose of a screen is to assess ‘impairment’ it could be argued that the inclusion of high item variance could contribute to false positive identification of impairment where there is lower-than-average premorbid intelligence.
Line 209. The authors state the Table 4 shows Pearson correlations between the domains and socio-demographic and injury factors, but then they describe only the correlations between the NUCOG domains. The authors should also describe the correlations between the domains and socio-demographic/injury factors. The authors may also want to do correlation analysis for total NUCOG score.
Response: We have added the following to lines 223-224
There were no significant associations between demographic variables and NUCOG domains.
Line 31. In the sentence: “…and its ability to…” it is not clear what the “its” refers to. Please clarify
Response: We have changed "its" to "the screens"
Reviewer 2 Report
“The need for a specialized neurocognitive screen and consistent cognitive impairment criteria in spinal cord injury: analysis of the suitability of the Neuropsychiatry Unit Cognitive Assessment Tool”
Overall strengths of the article:
Evidence indicates that spinal cord injury (SCI) is associated with accelerated cognitive aging, resulting in, slowed electroencephalographic brain activity and slowed processing speed. This manuscript aims to investigate the suitability of the Neuropsychiatry Unit Cognitive Assessment Tool (NUCOG) as a neurocognitive screen for adults with SCI by examining its sensitivity in detecting cognitive impairment in SCI individuals. The NUCOG total score was shown to measure cognitive capacity suitably in adults with SCI. Interestingly, NUCOG detected differences associated with time since injury. A Well detailed and executed paper examining the NUCOG for cognitive impairment analysis criteria in SCI. Overall, the methods are well detailed, the statistical analysis is rigorous, and the data well presented (except for maybe a few critiques detailed below).
Specific comments on weaknesses:
Major Critical Comments:
1. “Participants in rehabilitation were found to have lower cognitive scores compared to community participants. Lower cognitive scores in participants engaged in rehabilitation can be explained, at least partially, by increased stress”. I had a hard time understanding this, isn’t rehabilitation should decrease the overall stress?
Minor points:
1. A better short title suggestion; “Neuropsychiatry unit cognitive assessment tool for cognitive impairment analysis criteria in spinal cord injury”
2. I will suggest removing abbreviations from the abstract.
3. Page 1, Abstract-line 28; ‘SCI sample’, table 3 title, page 14 line 234. Likewise, on Page 4, line 76; the ‘able-bodied sample’ and the ‘SCI sample’ they are people, not samples, I think ‘subject’ is a better word.
4. Table 1 needs to be formatted.
Author Response
Evidence indicates that spinal cord injury (SCI) is associated with accelerated cognitive aging, resulting in, slowed electroencephalographic brain activity and slowed processing speed. This manuscript aims to investigate the suitability of the Neuropsychiatry Unit Cognitive Assessment Tool (NUCOG) as a neurocognitive screen for adults with SCI by examining its sensitivity in detecting cognitive impairment in SCI individuals. The NUCOG total score was shown to measure cognitive capacity suitably in adults with SCI. Interestingly, NUCOG detected differences associated with time since injury. A Well detailed and executed paper examining the NUCOG for cognitive impairment analysis criteria in SCI. Overall, the methods are well detailed, the statistical analysis is rigorous, and the data well presented (except for maybe a few critiques detailed below).
Response: Thank you for these kind words.
Specific comments on weaknesses:
Major Critical Comments:
“Participants in rehabilitation were found to have lower cognitive scores compared to community participants. Lower cognitive scores in participants engaged in rehabilitation can be explained, at least partially, by increased stress”. I had a hard time understanding this, isn’t rehabilitation should decrease the overall stress?
Response: To enhance understanding we have changed this to in line 235-237:
Lower cognitive scores in participants engaged in rehabilitation can might be explained, at least partially, by compromised attentional resources associated with anxiety increased stress/upheaval associated with the adversity of an acute SCI,
Minor points:
A better short title suggestion; “Neuropsychiatry unit cognitive assessment tool for cognitive impairment analysis criteria in spinal cord injury”
Response: Thank you for the suggestion, however while on the long side, we feel the title explains well the purpose of the study.
I will suggest removing abbreviations from the abstract.
Response: There are only 3 abbreviations: SCI, MCI and CFA. We don't think this distracts from the Abstract and removing abbreviations would add a substantial number of words and place pressure on keeping the word limit around 200 words
Page 1, Abstract-line 28; table 3 title, page 14; ‘SCI sample’,
Response: changed to "adults with SCI"
line 234. Response: "SCI sample" is removed
Likewise, on Page 4, line 76; the ‘able-bodied sample’
Response: changed to "sample of able-bodied adults"
and the ‘SCI sample’ they are people, not samples,
- Table 1 needs to be formatted. Response: Done
Round 2
Reviewer 1 Report
The authors have addressed all the comments adequately.